# A Hierarchy of Graph Neural Networks Based on Learnable Local Features

## Abstract

Graph neural networks (GNNs) are a powerful tool to learn representations on graphs by iteratively aggregating features from node neighbourhoods. Many variant models have been proposed, but there is limited understanding on both how to compare different architectures and how to construct GNNs systematically. Here, we propose a hierarchy of GNNs based on their aggregation regions. We derive theoretical results about the discriminative power and feature representation capabilities of each class. Then, we show how this framework can be utilized to systematically construct arbitrarily powerful GNNs. As an example, we construct a simple architecture that exceeds the expressiveness of the Weisfeiler-Lehman graph isomorphism test. We empirically validate our theory on both synthetic and real-world benchmarks, and demonstrate our example's theoretical power translates to strong results on node classification, graph classification, and graph regression tasks.

## 1 Introduction

Graphs arise naturally in the world and are key to applications in chemistry, social media, finance, and many other areas. Understanding graphs is important and learning graph representations is a key step. Recently, there has been an explosion of interest in utilizing graph neural networks (GNNs), which have shown outstanding performance across tasks (e.g. Kipf & Welling (2016), Veličković et al. (2017)). Generally, we consider node-feature GNNs which operate recursively to aggregate representations from a neighbouring region (Gilmer et al., 2017).

In this work, we propose a representational hierarchy of GNNs, and derive the discriminative power and feature representation capabilities in each class. Importantly, while most previous work has focused on GNNs aggregating over vertices in the immediate neighbourhood, we consider GNNs aggregating over arbitrary subgraphs containing the node. We show that, under mild conditions, there is only in fact a small class of subgraphs that are valid aggregation regions. These subgraphs provide a systematic way of defining a hierarchy for GNNs.

Using this hierarchy, we can derive theoretical results which provide insight into GNNs. For example, we show that no matter how many layers are added, networks which only aggregate over immediate neighbors cannot learn the number of triangles in a node's neighbourhood. We demonstrate that many popular frameworks, including GCN[1] (Kipf & Welling, 2016), GAT (Veličković et al., 2017), and N-GCN (Abu-El-Haija et al., 2018) are unified under our framework. We also compare each class using the Weisfeiler-Lehman (WL) isomorphism test (Weisfeiler & Lehman, 1968), and conclude our hierarchy is able to generate arbitrarily powerful GNNs. Then we utilize it to systematically generate GNNs exceeding the discriminating power of the 1-WL test.

Experiments utilize both synthetic datasets and standard GNN benchmarks. We show that the method is able to learn difficult graph properties where standard GCNs fail, even with multiple layers. On benchmark datasets, our proposed GNNs are able to achieve strong results on multiple datasets covering node classification, graph classification, and graph regression.

---

[1]Throughout the paper we use GCN to specifically refer to the model proposed in Kipf & Welling (2016).

## 2    RELATED WORK

Numerous works (see Li et al. (2015), Atwood & Towsley (2016), Defferrard et al. (2016), Kipf & Welling (2016), Niepert et al. (2016), Santoro et al. (2017), Veličković et al. (2017), Verma & Zhang (2018), Zhang et al. (2018a), Ivanov & Burnaev (2018), Wu et al. (2019a) for examples) have constructed different architectures to learn graph representations. Collectively, GNNs have pushed the state-of-the-art on many different tasks on graphs, including node classification, and graph classification/regression. However, there are relatively few works that attempt to understand or categorize GNNs theoretically.

Scarselli et al. (2009) presented one of the first works that investigated the capabilities of GNNs. They showed that the GNNs are able to approximate a large class of functions (those satisfying preservation of the unfolding equivalence) on graphs arbitrarily well. A recent work by Xu et al. (2018) also explored the theoretical properties of GNNs. Its definition of GNNs is limited to those that aggregate features in the immediate neighbourhood, and thus is a special case of our general framework. We also show that the paper's conclusion that GNNs are at most as powerful as the Weisfeiler-Lehman test fails to hold in a simple extension.

Survey works including Zhou et al. (2018) and Wu et al. (2019b) give an overview of the current field of research in GNNs, and provide structural classifications of GNNs. We differ in our motivation to categorize GNNs from a computational perspective. We also note that our classification only covers static node feature graphs, though extensions to more general settings are possible.

The disadvantages of GNNs using localized filter to propagate information are analyzed in Li et al. (2018). One major problem is their incapability of exploring global graph structures. To alleviate this, there has been two important works in expanding neighbourhoods: N-GCN (Abu-El-Haija et al., 2018) feeds higher-degree polynomials of adjacency matrix to multiple instantiations of GCNs, and Morris et al. (2018) generalizes GNNs to $k$-GNNs by constructing a set-based $k$-WL to consider higher-order neighbourhoods and capture information beyond node-level. We compare architectures constructed using our hierarchy to these previous works in the experiments, and show that systematic construction of higher-order neighbourhoods brings an advantage across different tasks.

## 3    BACKGROUND

Let $G = (V, E)$ denote an undirected and unweighted graph, where $|V| = N$, and $|E| = \Omega$. Unless otherwise specified, we include self-loops for every node $v \in V$. Let $A$ be the graph's adjacency matrix. Denote $d(u, v)$ as the distance between two nodes $u$ and $v$ on a graph, defined as the minimum length of walk between $u$ and $v$. We further write $d_v$ as the degree of node $v$, and $\mathcal{N}(v)$ as the set of nodes in the direct neighborhood of $v$ (including $v$ itself).

Graph Neural Networks (GNNs) utilize the structure of a graph $G$ and node features $X \in \mathbb{R}^{N \times p}$ to learn a refined representation of each node, where $p$ is input feature size, i.e. for each node $v \in V$, we have features $X_v \in \mathbb{R}^p$.

A GNN is a function that for every layer $l$ at every node $v$ aggregates features over a connected subgraph $G_v \subseteq G$ containing the node $v$, and updates a hidden representation $H^{(l)} = [h_1^{(l)}, \cdots, h_N^{(l)}]$. Formally, we can define the $l$th layer of a GNN (with $h_v^{(0)} = X_v$):

$$a_v^{(l)} = \text{Agg}\big|_{G_v}(H^{(l-1)}) \qquad h_v^{(l)} = \text{Com}(h_v^{(l-1)}, a_v^{(l)})$$

where $|$ is the restriction symbol over the domain $G_v$, the aggregation subgraph. The aggregation function $\text{Agg}(\cdot)$ is invariant with respect to the labeling of the nodes. The aggregation function, $\text{Agg}(\cdot)$, summarizes information from a neighbouring region $G_v$, while the combination function $\text{Com}(\cdot)$ joins such information with the previous hidden features to produce a new representation.

For different tasks, these GNNs are combined with an output layer to coerce the final output into an appropriate shape. Examples include fully-connected layers (Xu et al., 2018), convolutional layers (Zhang et al., 2018a), and simple summation (Verma & Zhang, 2018). These output layers are

task-dependent and not graph-dependent, so we would omit these in our framework, and consider the node level output $H^{(L)}$ of the final $L$th layer as the output of the GNN.

We consider three representative GNN variants in terms of this notation, where $W^{(l)}$ is a learnable weight matrix at layer $l$:[2]

- **Graph Convolutional Networks** (GCNs) (Kipf & Welling, 2016):
$$\text{Agg}(\cdot) = \sum_{u \in \mathcal{N}(v)} h_u^{(l-1)} \qquad \text{Com}(\cdot) = \text{ReLu}(a_v^{(l)} W^{(l)})$$

- **Graph Attention Networks** (GAT) (Veličković et al., 2017):
$$\text{Agg}(\cdot) = \sum_{u \in \mathcal{N}(v)} \text{softmax}_{u \in \mathcal{N}(v)} \left( \text{MLP}(h_u^{(l-1)}, h_v^{(l-1)}) \right) h_u^{(l-1)} \qquad \text{Com}(\cdot) = \text{ReLu}(a_v^{(l)} W^{(l)})$$

- **N-GCN** (Abu-El-Haija et al., 2018) (2-layer case):
$$\text{Agg}(\cdot) = \sum_{u_1 \in \mathcal{N}(v)} \sum_{u_2 \in \mathcal{N}(u_1)} h_{u_2}^{(l-1)} \qquad \text{Com}(\cdot) = \text{ReLu}(a_v^{(l)} W^{(l)})$$

## 4 Hierarchical Framework for Constructing GNNs

Our proposed framework uses walks to specify a hierarchy of aggregation ranges. The aggregation function over a node $v \in G$ is a permutation-invariant function over a connected subgraph $G_v$. Consider the simplest case, using the neighbouring vertices $u \in \mathcal{N}(v)$, utilized by many popular architectures (e.g. GCN, GAT). Then $G_v$ in this case is a star-shaped subgraph, as illustrated below in Figure 1. We refer to that as $D_1(v)$, which in terms of walks, is the union of all edges and nodes in length-2 walks that start and end at $v$.

To build a hierarchy, we consider benefits of longer walks. The next simplest graph feature is the triangles in the neighbourhood of $v$. Knowledge on connections between the neighbouring nodes of $v$ are necessary for considering triangles. A natural formulation using walks would be length-3 walks that start and end at $v$. A length-3 returning walk outlines a triangle, and the union of all length-3 returning walks induces a subgraph, formed by all nodes and edges included in those walks. This is illustrated in Figure 1 as $L_1(v)$.

**Definition 1.** *Define the set of all walks of length $\leq m$ returning to $v$ as $W_m(v)$. For $k \in \mathbb{Z}^+$, we define $D_k(v)$ as the subgraph formed by all the edges and nodes in $W_{2k}(v)$, while $L_k(v)$ is defined as the subgraph formed by all the nodes and edges in $W_{2k+1}(v)$.*

Intuitively, $L_k(v)$ is a subgraph of $G$ consisting of all nodes and edges in the $k$-hop neighbourhood of node $v$, and $D_k(v)$ only differs from $L_k(v)$ by excluding the edges between the distance-$k$ neighbors of $v$. We explore this further in Section 5. An example illustration of the neighbourhoods defined above is shown in Figure 1.

This set of subgraphs naturally induces a hierarchy with increasing aggregation region:

**Definition 2.** *The D-L hierarchy of aggregation regions for a node $v$, $\mathcal{A}_{D-L}(v)$ in a graph $G$ is, in increasing order:*
$$\mathcal{A}_{D-L}(v) = \{D_1(v), L_1(v), \cdots, D_k(v), L_k(v), \cdots\} \tag{1}$$
*Where $D_1(v) \subseteq L_1(v) \subseteq D_2(v) \subseteq L_2(v) \cdots$.*

Next, we consider the properties of this hierarchy. One important property is completeness - that the hierarchy can classify every possible GNN. Note that there is no meaningful complete hierarchy if $G_v$ is arbitrary. Therefore, we propose to limit our focus to those $G_v$ that can be defined as a function of the distance from $v$. Absent specific graph structures, distance is a canonical metric between vertices and this definition includes all examples listed in Section 3. With such assumption, we can show that the D-L hierarchy is complete:

---

[2]For simplicity we present the version without feature normalization using node degrees.

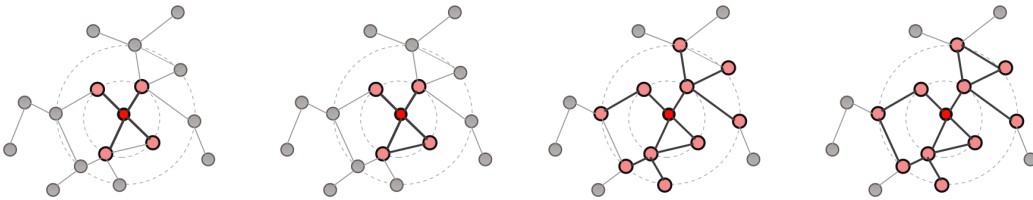

Figure 1: Illustration of D-L aggregation regions. Dashed circles represent neighborhoods of different hops. From left to right: $D_1$, $L_1$, $D_2$, and $L_2$. Both $D_k$ and $L_k$ include nodes within the k-hop neighborhood, but $D_k$ does not include edges between nodes on the outmost ring whereas $L_k$ does.

**Theorem 1.** *Consider a GNN defined by its action at each layer:*

$$a_v^{(l)} = \text{Agg}\big|_{G_v}(H^{(l-1)}) \qquad h_v^{(l)} = \text{Com}(h_v^{(l-1)}, a_v^{(l)}) \tag{2}$$

*Assume $G_v$ can be defined as a univariate function of the distance from $v$. Then both of the following statements are true for all $k \in \mathbb{Z}^+$:*

- *If $D_k(v) \subseteq G_v \subseteq L_k(v)$, then $G_v \in \{D_k(v), L_k(v)\}$.*

- *If $L_k(v) \subseteq G_v \subseteq D_{k+1}(v)$, then $G_v \in \{L_k(v), D_{k+1}(v)\}$.*

This theorem shows that one cannot create an aggregation region based on node distance that is "in between" the hierarchy defined. With Theorem 1, we can use the D-L aggregation hierarchy to create a hierarchy of GNNs based on their aggregation regions.

**Definition 3.** *For $k \in \mathbb{Z}^+$, $\mathcal{G}(D_k)$ is the set of all graph neural networks with aggregation region $G_v = D_k(v)$ that is not a member of $\mathcal{G}(L_k)$. $\mathcal{G}(L_k)$ is the set of all graph neural networks with aggregation region $G_v = L_k(v)$ that is not a member of $\mathcal{G}(D_{k-1})$.*

We explicitly exclude those belonging to a lower aggregation region in order to make the hierarchy well-defined (otherwise a GNN of order $\mathcal{G}(D_1)$ is trivially one of order $\mathcal{G}(L_1)$). We also implicitly define $D_0 = L_0 = \emptyset$.

## 4.1 CONSTRUCTING D-L GNNS

The D-L Hierarchy can be used both to classify existing GNNs and also to construct new models. We first note that all GNNs which aggregate over immediate neighbouring nodes fall in the class of $\mathcal{G}(D_1)$. For example, Graph Convolutional Networks (GCNs) defined in Section 3 is in $\mathcal{G}(D_1)$ since its aggregation is $\text{Agg}(\cdot) = \sum_{u \in D_1(v)} h_u^{(l-1)}$, and similarly the N-GCN example is in $\mathcal{G}(D_2)$. Note that these classes are defined by the subgraph used by $\text{Agg}$, but does not imply that these networks reach the maximum discriminatory power of their class (defined in the next section).

We can use basic building blocks to implement different levels of GNNs. These examples are not meant to be exhaustive and only serve as a glimpse of what could be achieved with this framework.

**Examples.** *For every $k \in \mathbb{Z}^+$:*

- *Any GNN with $\text{Agg}(\cdot) = \displaystyle\sum_{u \in D_k(v)} (A^k)_{vu} h_u^{(l-1)}$ is a GNN of class $\mathcal{G}(D_k)$.*

- *Any GNN with $\text{Agg}(\cdot) = \displaystyle\sum_{u \in D_k(v)} (A^{k+1})_{vu} h_u^{(l-1)}$ is a GNN of class $\mathcal{G}(L_k)$.*

- *Any GNN with* $\mathrm{Agg}(\cdot) = (A^{2k+1})_{vv} \cdot h_v^{(l-1)}$ *is a GNN of class* $\mathcal{G}(L_k)$.

Intuitively, $(A^k)_{vu}$ counts all $k$-length walks from $v$ to $u$, which includes all nodes in the $k$-hop neighbourhood. The difference between the first and the second example above is that in the second one, we allow $(k+1)$-length walks from the nodes in the $k$-hop neighbourhood, which promotes it to be class of $\mathcal{G}(L_k)$. Note the simplicity of the first and the last examples: in matrix form the first is $A^k H^{(l-1)}$ while the last form is $\mathrm{Diag}(A^{2k+1}) \cdot H^{(l-1)}$.

The building blocks can be gradually added to the original aggregation function. This is particularly useful if an experimenter knows there are higher-level properties that are necessary to compute, for instance to incorporate knowledge of triangles, one can design the following network (see Section 6 for more details):

$$w_1 A H^{(l-1)} + w_2 \mathrm{Diag}(A^3) \cdot H^{(l-1)} \tag{3}$$

where $w_1, w_2 \in \mathbb{R}$ are learnable weights.

## 5 THEORETICAL PROPERTIES

We can prove interesting theoretical properties for each class of graph neural networks on this hierarchy. To do this, we utilize the Weisfeiler-Lehman test, a powerful classical algorithm used to discriminate between potentially isomorphic graphs. In interest of brevity, its introduction is included in the Appendix in Section 8.1.

We define the terminology of "discriminating graphs" formally below:

**Definition 4.** *The discriminative power of a function $f$ over graphs $G$ is the set of graphs $S_G^f$ such that for every pair of graphs $G_1, G_2 \in S_G^f$, the function has $f(G_1) = f(G_2)$ iff $G_1 \cong G_2$ and $f(G_1) \neq f(G_2)$ iff $G_1 \ncong G_2$. We say $f$ decides $G_1, G_2$ as isomorphic if $f(G_1) = f(G_2)$ and vice versa.*

Essentially, $S_G^f$ is the set of graphs that $f$ can decide correctly whether any two of them are isomorphic or not. We say $f$ has a *greater* discriminative power than $g$ if $S_G^f \supsetneq S_G^g$. Now we first introduce a theorem proven by Xu et al. (2018):

**Theorem 2.** *The maximum discriminative power of the set of GNNs in $\mathcal{G}(D_1)$ is strictly less than or equal to the 1-dimensional WL test.*

Their framework only included $\mathcal{G}(D_1)$ GNNs, and they upper bounded the discriminative power of such GNNs. With our generalized framework, we are able to prove a slightly surprising result:

**Theorem 3.** *The maximum discriminative power of the set of GNNs in $\mathcal{G}(L_1)$ is strictly greater than the 1-dimensional WL test.*

This result is central to understanding GNNs. Even though the discriminative power of $\mathcal{G}(D_1)$ is strictly less than or equal to the 1-WL test, Theorem 3 shows that just by adding the connections between the immediate neighbors of each node ($L_1 \backslash D_1$), we can achieve theoretically greater discriminative power.

One particular implication is that GNNs with maximal discriminative power in $\mathcal{G}(L_1)$ can count the number of triangles in a graph, while those in $\mathcal{G}(D_1)$ cannot, no matter how many layers are added. This goes against the intuition that more layers allow GNNs to aggregate information from further nodes, as $\mathcal{G}(D_1)$ is unable to aggregate the information of triangles from the $L_1$ region, which is important in many applications (see Frank & Strauss (1986), Tsourakakis et al. (2011), Becchetti et al. (2008), Eckmann & Moses (2002)).

Unfortunately, this is the only positive result we are able to establish regarding the WL test as the $k$-dim WL-test is not a local method for $k > 1$. Nevertheless, we are able to prove that our hierarchy admits arbitrarily powerful GNNs through the following theorem:

| GNN Class | Computational Complexity | Maximum Discriminatory Power | Possible Learned Features |
|-----------|--------------------------|------------------------------|---------------------------|
| $\mathcal{G}(D_1)$ | $\leq O(\Omega p)$ | $\leq$1-WL | Node Degree |
| $\mathcal{G}(L_1)$ | $\leq O(\Omega N + Np)$ | >1-WL 
 All graphs of $\leq 2$ nodes | All Cliques 
 Length 3 cycles (Triangles) |
| $\mathcal{G}(D_2)$ | $\leq O(\Omega p)$ | >1-WL 
 All graphs of $\leq 2$ nodes | Length 2 walks 
 Length 4 cycles |
| $\mathcal{G}(D_k)$ | $\leq O(k\Omega p)$ | >1-WL 
 All graphs of $\leq k$ nodes | Length $k$ walks 
 Length $2k$ cycles |
| $\mathcal{G}(L_k)$ | $\leq O(k\Omega N + Np)$ | >1-WL 
 All graphs of $\leq k + 1$ nodes | Length $2k + 1$ cycles |

Table 1: Properties of different GNN classes. Shows the upper bound computational complexity when the maximum discriminatory power is obtained. Here we assume hidden size $p$ is the same as feature input size. Final column contains some examples of features that can be learned by each class.

**Theorem 4.** *For all $k \in \mathbb{Z}^+$, there exists a GNN within the class of $\mathcal{G}(L_k)$ that is able to discriminate all graphs with $\leq k + 1$ nodes.*

This shows that as $k \to \infty$, we are able to discriminate all graphs. We record the full set of results proven in Table 1. The key ingredients for proving these results are contained in Appendix 8.3 and 8.4. Here we see that at the $\mathcal{G}(L_1)$ class, theoretically we are able to learn all cliques (as cliques by definition are fully connected). As we gradually move upward in the hierarchy, we are able to learn more far-reaching features such as higher length walks and cycles, while the discriminatory power improves. We also note that the theoretical complexity increases as $k$ increases.

## 6 EXPERIMENTS

We consider the capability of two specific GNNs instantiations that are motivated by this framework: $w_1 A H^{(l-1)} + w_2 \text{Diag}(A^3) \cdot H^{(l-1)}$ (GCN-L1) and $w_1 A H^{(l-1)} + w_2 \text{Diag}(A^3) \cdot H^{(l-1)} + w_3 A^2 H^{(l-1)}$ (GCN-D2). These can be seen as extensions of the GCN introduced in Kipf & Welling (2016). The first, GCN-L1, equips the GNN with the ability to count triangles. The second, GCN-D2, can further count the number of 4-cycles. We note their theoretical power below (proof follows from Theorem 3):

**Corollary 1.** *The maximum discriminative power of GCN-L1 and GCN-D2 is strictly greater than the 1-dimensional WL test.*

We compare the performance of GCN-L1, GCN-D2 and other state-of-art GNN variants on both synthetic and real-world tasks.[3] For the combine function of GCN, GCN-L1, and GCN-D2, we use $\text{Com}(\cdot) = \text{MLP}(a_v^{(k)})$, where MLP is a multi-layer perceptron (MLP) with LeakyReLu activation similar to Xu et al. (2018).

All of our experiments are run with PyTorch 1.2.0, PyTorch-Geometric 1.2.1, and we use NVIDIA Tesla P100 GPUs with 16GB memory.

### 6.1 SYNTHETIC EXPERIMENTS

To verify our previous claim that in our proposed hierarchy, GNNs from certain classes are able to learn specific features more effectively, we created two tasks: predict the number of triangles and number of 4-cycles in the graphs. For each task, the dataset contains 1000 graphs and is generated in a procedure as follows: We fix the number of nodes in each graph to be 100 and use the Erdős–Rényi random graph model to generate random graphs with edge probability 0.07. Then we count the number of patterns of interest. In the 4-cycle dataset, the average number of 4-cycles in each graph is 1350 and in the triangle dataset, there are 54 triangles on average in each graph.

---

[3]The code is available at a public github repository. Reviewers have anonymized access through supplementary materials. The synthetic datasets are included in the codebase as well.

|  | **MSE # Triangles** | **MSE # 4 Cycles** ($\times 10^3$) |
|---|---|---|
| Predict Mean (Baseline) | $125.4 \pm 10.7$ | $36.4 \pm 5.6$ |
| GCN (2-layer) | $506.2 \pm 80.9$ | $142.3 \pm 19.8$ |
| GCN (3-layer) | $485.0 \pm 92.4$ | $136.7 \pm 18.5$ |
| GCN-L1 (1-layer) | $\mathbf{61.2 \pm 11.6}$ | $45.2 \pm 6.0$ |
| GCN-D2 (1-layer) | $\mathbf{57.9 \pm 18.0}$ | $\mathbf{3.0 \pm 1.0}$ |

Table 2: Results of experiments on synthetic datasets (i) Count the number of triangles in the graph (ii) Count the number of 4 cycles in the graph. The reported metric is MSE over the testing set.

| **Dataset** | **Category** | **# Graphs** | **# Classes** | **# Nodes Avg.** | **# Edges Avg** | **Task** |
|---|---|---|---|---|---|---|
| Cora (Yang et al., 2016) | Citation | 1 | 7 | 2,708 | 5,429 | NC |
| Citeseer (Yang et al., 2016) | Citation | 1 | 6 | 3,327 | 4,732 | NC |
| PubMed (Yang et al., 2016) | Citation | 1 | 3 | 19,717 | 44,338 | NC |
| NCI1 (Shervashidze et al., 2011) | Bio | 4,110 | 2 | 29.87 | 32.30 | GC |
| Proteins (Kersting et al., 2016) | Bio | 1,113 | 2 | 39.06 | 72.82 | GC |
| PTC-MR (Kersting et al., 2016) | Bio | 344 | 2 | 14.29 | 14.69 | GC |
| MUTAG (Borgwardt et al., 2005) | Bio | 188 | 2 | 17.93 | 19.79 | GC |
| QM7b (Wu et al., 2018) | Bio | 7,210 | 14 | 16.42 | 244.95 | GR |
| QM9 (Wu et al., 2018) | Bio | 133,246 | 12 | 18.26 | 37.73 | GR |

Table 3: Details of benchmark datasets used. Types of tasks are: NC for node classification, GC for graph classification, GR for graph regression.

We perform 10-fold cross-validation and record the average and standard deviation of evaluation metrics across the 10 folds within the cross-validation. We used 16 hidden features, and trained the networks using Adam optimizer with 0.001 initial learning rate, $L_2$ regularization $\lambda = 0.0005$. We further apply early stopping on validation loss with a delay window size of 10. The dropout rate is 0.1. The learning rate is scheduled to reduce $50\%$ if the validation accuracy stops increasing for 10 epochs. We utilized a two-layer MLP in our combine function for GCN, GCN-L1 and GCN-L2, similar to the implementation in Xu et al. (2018). For training stability, we limited $w_1, w_2 \in (0, 1)$ in our models using the sigmoid function.

**Results** In our testing, we limited GCN-L1 and GCN-D2 to a 1-layer network. This prevents GCN-L1 and GCN-D2 from utilizing higher order features to reverse predict the triangles and 4-cycles. Simultaneously, we ensured GCN had the same receptive field as such networks by using 2-layer and 3-layer GCNs, which provided GCN with additional feature representational capability. The baseline is a model that predicts the training mean on the testing set. The results are in Table 2. GCN completely fails to learn the features (worse than a simple mean prediction). However, we see that GCN-L1 effectively learns the triangle counts and greatly outperforms the mean baseline, while GCN-D2 is similarly able in providing a good approximation on the count of 4-cycles, without losing the ability to count triangles. This validates the "possible features learned" in Table 1.

## 6.2 REAL-WORLD BENCHMARKS

We next consider standard benchmark datasets for (i) node classification, (ii) graph classification, (iii) graph regression tasks. The details of these datasets are presented in Table 3.

The setup of the learning rate scheduling and $L_2$ regularization rate are the same as in synthetic tasks. For the citation tasks, we used 16 hidden features, while we used 64 for the biological datasets. Since our main modification is the expansion of aggregation region, our main comparison benchmarks are $k$-GNN (Morris et al., 2018) and N-GCN (Abu-El-Haija et al., 2018), two previous best attempts in incorporating aggregation regions beyond the immediate nodes. Note that we can view a $m$th order N-GCN as aggregating over $D_1, D_2, \cdots, D_m$.

We further include results on GAT (Verma & Zhang, 2018), GIN (Xu et al., 2018), RetGK (Zhang et al., 2018b), GNTK (Du et al., 2019), WL-subtree (Shervashidze et al., 2011), SHT-PATH, (Borgwardt & Kriegel, 2005) and PATCHYSAN (Niepert et al., 2016) to show some best performing architectures.

| Dataset | Cora | Citeseer | PubMed | NCI1 | Proteins | PTC-MR | MUTAG | QM7b | QM9 |
|---|---|---|---|---|---|---|---|---|---|
| GAT | **83.0 ± 0.7** | **72.5 ± 0.7** | 79.0 ± 0.3 | 74.5 ± 3.5* | 73.7 ± 5.6* | 60.2 ± 3.0* | 84.0 ± 8.0* | 91.7 ± 5.5* | 115.0 ± 17.5* |
| GIN | 77.6 ± 1.1 | 66.1 ± 0.9 | 77.0 ± 1.2 | 82.7 ± 1.7 | **76.2 ± 2.8** | **64.6 ± 7.0** | 89.4 ± 8.6 | | |
| WL-OA | | | | **86.1** | 75.3 | **63.6** | 84.5 | | |
| P.SAN | | | | 78.5 ± 1.8 | 75.8 ± 2.7 | 60.0 ± 4.8 | **92.6 ± 4.2** | | |
| RetGK | | | | **84.5 ± 0.2** | 75.8 ± 0.6 | 62.5 ± 1.6 | **90.3 ± 1.1** | | |
| GNTK | | | | **84.2 ± 1.2** | 75.6 ± 4.2 | **67.9 ± 6.9** | 90.0 ± 8.5 | | |
| S-PATH | | | | 73.0 ± 0.5 | 75.0 ± 0.5 | 58.5 ± 2.5 | 85.7 ± 2.5 | | |
| N-GCN | **83.0** | **72.2** | 79.5 | 75.8 ± 1.9* | **76.5 ± 1.5*** | 61.0 ± 5.0* | 85.0 ± 6.9* | 82.0 ± 5.4* | 120.7 ± 8.5* |
| k-GNN | 81.6 ± 0.4* | 71.5 ± 0.5* | **79.8 ± 0.3*** | 76.2 | 75.5 | 60.9 | 86.1 | 75.1 ± 8.5* | 104.2 ± 10.4 |
| GCN | 80.6 ± 1.4 | 70.3 ± 1.2 | 79.0 ± 0.4 | 73.2 ± 1.4 | 73.9 ± 2.8 | 59.0 ± 2.0 | 82.2 ± 5.1 | 104.3 ± 15.6 | 160.2 ± 15.4 |
| GCN-L1 | **82.5 ± 0.3** | **72.0 ± 0.3** | **80.2 ± 0.2** | 79.5 ± 1.6 | **77.6 ± 3.8** | **64.1 ± 2.5** | 86.8 ± 8.3 | **52.4 ± 4.3** | **78.5 ± 8.6** |
| GCN-D2 | **82.9 ± 1.0** | **72.3 ± 0.3** | **80.2 ± 0.3** | 77.0 ± 2.0 | **77.0 ± 3.0** | 64.6 ± 4.1 | 87.8 ± 5.6 | 49.0 ± 2.9 | 72.5 ± 13.0 |

Table 4: Results of experiments on real-world datasets. The reported metrics are accuracy on classification tasks and MSE on regression tasks. Figures for comparative methods are from literature except for those with *, which come from our own implementation. The best-performing architectures are highlighted in bold.

Baseline neural network models use a 1-layer perceptron combine function, with the exception of $k$-GNN, which uses a 2-layer perceptron combine function. Thus, to illustrate the effectiveness of the framework, we only utilize a 1-layer perceptron combine function for all tasks for our GCN models, with the exception of NCI1. 2-layer perceptrons seemed necessary for good performance in NCI1, and thus we implemented all neural networks with 2-layer perceptrons for this task to ensure a fair comparison. We tuned the learning rates $\in \{0.001, 0.005, 0.01, 0.05\}$ and dropout rates $\in \{0.1, 0.5\}$. For numerical stability, we normalize the aggregation function using the degree of $v$ only. For the node classification tasks, we directly utilized the final layer output, while we summed over the node representations for the graph-level tasks.

**Results** Experimental results on real-world data are summarized in Table 4. According to our experiments, GCN-L1 and GCN-D2 noticeably improve upon GCN across all datasets, due to its ability to combine node features in more complex and nonlinear ways. The improvement is statistically significant on the $5\%$ level for all datasets except Proteins. The results of GCN-L1 and GCN-D2 match the best performing architectures in most datasets, and lead in numerical averages for Cora, Proteins, QM7b, and QM9 (though not statistically significant for all).

Importantly, the results also show a significant improvement from the two main comparison architectures, $k$-GNN and N-GCN. We see that further expanding aggregation regions generates diminishing returns on these datasets, and the majority of the benefit is gained in the first-order extension $\mathcal{G}(L_1)$. This is in contrast to N-GCN which skipped $L_1$ to only used $D$-type aggregation regions $(D_2, D_3, \cdots)$, which is an incomplete hierarchy of aggregation regions. The differential in results illustrates the power of the complete hierarchy as proven in Theorem 1.

We especially would like to stress the outsized improvement of GCN-L1 on the biological datasets. As described in 1, GCN-L1 is able to capture information about triangles, which are highly relevant for the properties of biological molecules. The experimental results verify such intuition, and show how knowledge about the task can lead to targeted GNN design using our framework.

## 7 CONCLUSION

We propose a theoretical framework to classify GNNs by their aggregation region and discriminative power, proving that the presented framework defines a complete hierarchy for GNNs. We also provide methods to construct powerful GNN models of any class with various building blocks. Our experimental results show that example models constructed in the proposed way can effectively learn the corresponding features exceeding the capability of 1-WL algorithm in graphs. Aligning with our theoretical analysis, experimental results show that these stronger GNNs can better represent the complex properties of a number of real-world graphs.

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

# 8 APPENDIXES

## 8.1 INTRODUCTION TO WEISFEILER-LEHMAN TEST

The 1-dimensional Weisfeiler-Lehman (WL) test is an iterative vertex classification method widely used in checking graph isomorphism. In the first iteration, the vertices are labeled by their valences. Then at each following step, the labels of vertices are updated by the multiset of the labels of themselves and their neighbors. The algorithm terminates when a stable set of labels is reached. The details of 1-dimensional Weisfeiler-Lehman (WL) is shown in Algorithm 1. Regarding the limitation of 1-dimensional Weisfeiler-Lehman (WL), Cai et al. (1992) described families of non-isomorphic graphs which 1-dimensional WL test cannot distinguish.

---

**Algorithm 1** 1-dimensional Weisfeiler-Lehman (WL)

---
1: **Input:** $G = (V, E)$, $G' = (V', E')$, initial labels $l_0(v)$ for all $v \in V \cup V'$
2: **Output:** stablized labels $l(v)$ for all $v \in V$
3: **while** $l_i(v)$ has not converged **do**          $\triangleright$ Until the labels reach stabalization
4:     **for** $v \in V \cup V'$ **do**
5:         $M_i(v) = $ multi-set $\{l_{i-1}(u) | u \in \mathcal{N}(v)\}$
6:     **end for**
7:     Sort $M_i(v)$ and concatenate them into string $s_i(v)$
8:     $l_i(v) = f(s_i(v))$, where $f$ is any function s.t. $f(s_i(v) = f(s_i(w)) \iff s_i(v) = s_i(w)$
9: **end while**

---

The k-dimensional Weisfeiler-Lehman (WL) algorithm extends the above procedure from operations on nodes to operations on tuples $V(G)^k$.

## 8.2 Proof of Theorem 1

**Theorem.** *Consider a GNN defined by its action at each layer:*

$$a_v^{(l)} = \text{Agg}\big|_{G_v}(H^{(l-1)}) \qquad h_v^l = \text{Com}(h_v^{(l-1)}, a_v^{(l)}) \tag{4}$$

*Assume $G_v$ can be defined as a univariate function of the distance from $v$. Then both of the following statements are true for all $k \in \mathbb{Z}^+$:*

- *If $D_k(v) \subseteq G_v \subseteq L_k(v)$, then $G_v \in \{D_k(v), L_k(v)\}$.*

- *If $L_k(v) \subseteq G_v \subseteq D_{k+1}(v)$, then $G_v \in \{D_{k+1}(v), L_k(v)\}$.*

*Proof.* We would prove by contradiction. Assume, in contrary, that one of the statements in Theorem 1 is false. Let $k \in \mathbb{Z}^+$. Then we would separate these two cases as below:

$\boldsymbol{D_k(v) \subsetneq G_v \subsetneq L_k(v)}$  Assume that $G_v$ satisfies such relationship. Since $L_k(v)$ and $D_k(v)$ only differ by the set $C_k(v) = \{e_{ij} \in E \mid d(i, v) = d(j, v) = k\}$, $G_v$ can only contain this set partially. Let $M_k(v) \subsetneq C_k(v)$ be the non-empty maximal subset of $C_k(v)$ that is contained in $G_v$. Since $M_k(v) \neq C_k(v)$, there exists $m, n$ with $d(v, m) = d(v, n) = k$ such that $e_{mn} \in C_k(v)$ but $e_{mn} \notin M_k(v)$. Consider a non-identity permutation of vertices fixing $v$. Then since $G_v$ is defined only using the distance function, and it needs to be permutation invariant, all $e_{ij}$ with $d(i, v) = d(j, v) = k$ must be in $C_k(v)$ and not in $M_k(v)$. But then $M_k(v)$ is empty, a contradiction.

$\boldsymbol{L_k(v) \subsetneq G_v \subsetneq D_{k+1}(v)}$  Assume that $G_v$ satisfies such relationship. Consider the set difference between $D_{k+1}(v)$ and $L_k(v)$, denoted as a subgraph $C_k(v)$:

$$C_k(v) = \big(V_k^C(v), E_k^C(v)\big) = (\{u \mid d(u, v) = k\}, \{e_{ij} \mid (d(v, i), d(v, j)) \in \{(k, k+1), (k+1, k)\}\})$$
$$\tag{5}$$

Then $G_v$ can only contain this set partially. Let $M_k(v) \subsetneq C_k(v)$ be the maximal subset of $C_k(v)$ that is contained in $G_v$. Since $M_k(v) \neq C_k(v)$, at least one of the followings must be true:

- There exists $u$ with $d(u, v) = k$ such that $u \in C_k(v)$ but $u \notin M_k(v)$.

- There exists $m, n$ with $d(v, m) = k, d(v, n) = k + 1$ such that $e_{mn} \in C_k(v)$ but $e_{mn} \notin M_k(v)$

For the first case, consider a non-identity permutation of the vertices fixing $v$. Then since $G_v$ is permutation invariant and defined only using the distance, then thus all vertices $w$ with $d(w, v) = k$ are in $C_k(v)$ but not in $M_k(v)$. This implies $V_k^M(v)$ is empty.

Using the same logic for the second case, one can conclude that all $e_{ij}$ with $d(v, i) = k$ and $d(v, m) = k + 1$ must be in $C_k(v)$ but not in $M_k(v)$. That means $E_k^M(v)$ is empty.

Therefore, we can conclude that at least one of $V_k^M(v)$ and $E_k^M(v)$ is empty. Since both cannot be empty (as that means $G_v = L_k(v)$), we must have either $V_k^M(v) = \emptyset$ or $E_k^M(v) = \emptyset$. With the former case $M_k(v)$ is not a valid subgraph (as some edges to nodes with distance $k + 1$ from $v$ are in the set, but the nodes are not), and with the latter case it is not connected (as the nodes with distance $k + 1$ from $v$ are in the set but none of the edges are), so neither of them are valid aggregation regions in our framework (our definition of GNN requires the region to be a connected graph). Thus, we reach a contradiction. $\square$

### 8.3 PROOF OF THEOREM 3

**Theorem.** *The maximum discriminative power of the set of GNNs in $\mathcal{G}(L_1)$ is strictly greater than the 1-dimensional WL test.*

*Proof.* We first note that a GNN with the aggregation function (in matrix notation):

$$\text{Agg}(\cdot) = \text{Diag}(A^3) \cdot H^{(k)} \tag{6}$$

is a $\mathcal{G}(L_1)$-class GNN. Then note that $(A^3)_{ii}$ is the number of length 3 walks that start and end at $i$. These walks must be simple 3-cycles, and thus $(A^3)_{ii}$ is twice the number of triangles that contains $i$ (since for a triangle $\{i, j, k\}$, there would be two walks $i \to j \to k \to i$ and $i \to k \to j \to i$). Fürer (2017) showed that 1-WL test cannot measure the number of triangles in a graph (while 2-WL test can), so there exist graphs $G_1$ and $G_2$ such that 1-WL test cannot differentiate but the GNN above can due to different number of triangles in these two graphs (an example are two regular graphs with the same degree and same number of nodes).

Now Xu et al. (2018) proved that $\mathcal{G}(D_1)$ has a maximum discriminatory power of 1-WL, and since $L_1 \supsetneq D_1$, the maximum discriminatory power of $\mathcal{G}(L_1)$ is at least as great than that of $\mathcal{G}(D_1)$, which is 1-WL.

Thus combining these two results give the required theorem. □

We here note the computational complexity of $A^3$ using naive matrix multiplication requires $O(N^3)$ multiplications. However, by exploiting the sparsity and binary nature of $A$, there exist algorithms that can calculate $A^k$ with $O(\Omega N)$ additions (Razzaque et al. (2008)), and we thus derive a more favorable bound.

### 8.4 PROOF OF THEOREM 4

**Theorem.** *For all $k \in \mathbb{Z}^+$, there exists a GNN within the class of $\mathcal{G}(L_k)$ that is able to discriminate all graphs with $k + 1$ nodes.*

*Proof.* We would prove by induction on $k$. The base case for $k = 1$ is simple.

Assume the statement is true for all $k \leq k' - 1$. Let us prove the case for $k = k' \geq 2$.

We would separate the proof into three cases:

- $G_1$ and $G_2$ are both disconnected.

- One of $G_1$, $G_2$ is disconnected.

- $G_1$ and $G_2$ are both connected.

If we can create appropriate GNNs to discriminate graphs in each of the three cases (say $f_1, f_2, f_3$), then the concatenated function $f := [f_1, f_2, f_3]$ can discriminate all graphs. Therefore, we would prove the induction step separately for the three cases below.

#### 8.4.1 $G_1$ AND $G_2$ DISCONNECTED

We would use the Graph Reconstruction Conjecture as proved in the disconnected case (Harary, 1974):

**Lemma 1.** *For all $k \geq 2$, two disconnected graphs $G_1, G_2$ with $|V_1| = |V_2| = k$ are isomorphic if and only if the set of $k - 1$ sized subgraphs for these two graphs are isomorphic.*

Let $G_1$ and $G_2$ be any two disconnected graphs with $k' + 1$ nodes. By the induction assumption, there exist a GNN in $\mathcal{G}(L_{k'})$ such that it discriminates all graphs with $k'$ nodes. Denote that $f_{k'} : \mathcal{G} \to \mathbb{R}^{k' \times p}$.

Then by Lemma 1 above, we know that $G_1$ and $G_2$ are isomorphic iff:

$$\{f_{k'}(G_1^1), \cdots f_{k'}(G_1^{k'+1})\} \cong \{f_{k'}(G_2^1), \cdots f_{k'}(G_2^{k'+1})\} \tag{7}$$

Where $G_1^1, \cdots G_1^{k'+1}$ are the $k' + 1$ subgraphs of $G_1$ with size $k'$, and similarly for $G_2$. Then we define:

$$f_{k'+1}^i(G) = \sum_{i=1}^{k'+1} (f_{k'}(G^i))^i \qquad i = 1, \cdots k' + 1$$

$$f_{k'+1}(G) = [f_{k'+1}^1(G), \cdots, f_{k'+1}^{k'+1}(G)]$$

Then, by Theorem 4.3 in Wagstaff et al. (2019), $f_{k'+1}(G)$ is an injective, permutation-invariant function on $\{f_{k'}(G^1), \cdots f_{k'}(G^{k'+1})\}$. Therefore, $f_{k'+1}(G)$ is a GNN with an aggregation region of $L_{k'+1}$ that can discriminate $G_1$ and $G_2$. Thus, the induction step is proven.

### 8.4.2 $G_1$ AND $G_2$ CONNECTED

In the case where both $G_1$ and $G_2$ are connected, let $v_1, u_1$ be two vertices from each of the two graphs. Note that since $G_1$ and $G_2$ have $k' + 1$ nodes, by definition of $L_{k'}$, we have $L_{k'}(v_1) = G_1$ and $L_{k'}(u_1) = G_2$. Since $v_1, u_1$ are arbitrary, every node in the two graphs has an aggregation region of its entire graph. Then we define the Aggregation function as followed:

$$\mathrm{Agg}(v) = \mathrm{LO}(A_{L_{k'}(v)})$$

Where $A_{L_{k'}}$ is the adjacency matrix restricted to the $L_{k'}(v)$ subgraph, and $\mathrm{LO}(A)$ returns the lexicographical smallest ordering of the adjacency matrix as a row-major vector among all isomorphic permutations (where nodes are relabeled).[4] For example, take:

$$A = \begin{pmatrix} 1 & 1 \\ 1 & 0 \end{pmatrix}$$

There are two isomorphic permutations of this adjacency matrix, which are:

$$\begin{pmatrix} 1 & 1 \\ 1 & 0 \end{pmatrix} \qquad \begin{pmatrix} 0 & 1 \\ 1 & 1 \end{pmatrix}$$

The row-major vectors of these two adjacency matrices are $[1, 1, 1, 0]$ and $[0, 1, 1, 1]$, in which $[0, 1, 1, 1]$ is lexicographical smaller, so $\mathrm{LO}(A) = [0, 1, 1, 1]$. Note that this function is permutation invariant.

Then for $G_1$ and $G_2$ connected, $A_{L_{k'}(v)}$ is always the adjacency matrix of the full graph. Therefore if $G_1$ and $G_2$ are connected and isomorphic, then their adjacency matrices are permutations of each other, and thus their lexicographical smallest ordering of the row-major vector form of the adjacency matrix are identical. The converse is also clearly true as the adjacency matrix determines the graph.

Therefore, this function discriminates all connected graphs of $k' + 1$ nodes, and the induction step is proven.

### 8.4.3 $G_1$ DISCONNECTED AND $G_2$ CONNECTED

In the case where $G_1$ is disconnected and $G_2$ is connected, we define the aggregation function as the number of vertices in $L_{k'}(v)$, denoted $|V_{L_{k'}(v)}|$:

$$\mathrm{Agg}(v) = |V_{L_{k'}(v)}|$$

---

[4]Strictly speaking, for consistency of the output length, we would require padding to a length of $(k' + 1)^2$ if the adjacency matrix restricted to the $L_{k'}(v)$ subgraph is not the full adjacency matrix. However, since we only care about the behavior of the function when $G_1$ and $G_2$ are both connected, this scenario never happens, so it is not material for any part of the proof below.

This is a permutation-invariant function. Note that for the connected graph $G_2$ and any vertex $v$ in $G_2$, this function returns $\text{Agg}(v) = k' + 1$ as $L_{k'}(v) = G_2$. Therefore, every node has the same embedding $k' + 1$.

On the other hand, for the disconnected graph $G_1$, let $[U_1, \cdots, U_m]$ be the connected components of $G_1$. Then for a vertex $u \in U_i$, it is clear that $L_{k'}(u) = U_i$, and thus $\text{Agg}(u) = |V_{U_i}|$ for all $u \in U_i$. And since $|V_{U_i}| < k' + 1$ by construction, $\text{Agg}(u) < k' + 1$ for all $u \in U_i$, so the embedding of $G_1$ and $G_2$ are never equal when $G_1$ is connected and $G_2$ is disconnected.

Therefore, this function discriminate all graphs of $k' + 1$ nodes in which one is connected and one is disconnected, so the induction step is proven.

$\square$

