# OpenReview forum: "A Hierarchy of Graph Neural Networks Based on Learnable Local Features"
_ICLR.cc/2020/Conference — Reject_

### Official Review · AnonReviewer3 · 2019-10-23
**Official Blind Review #3**

**Rating:** 3

**Review:**

The paper proposes a generalized framework for GNN. It proposes a representational hierarchy of GNNs. (D-1, L-1,..D-n, L-n). L-k is the k-hop neighborhood including all edges. And D-k is L-k without edges between the outermost nodes. The discriminative power of a network using L-1 is > WL-1. On various graph classification/regression tasks, the proposed method shows good performance. Show promising result on QM7b QM9 graph regression task  (counting triangles, cycles etc) which are highly relevant to the proposed method.

Some concerns:

1, the theoretical results seem a bit incremental compared with (Xu et al. 2018).

2. it would be nice to comment on how this will affect cases with nontrivial node features and general node classification tasks.

3.  the empirical results are not very convincing. On standard datasets/tasks, the baselines are not state-of-the-arts. The results only show the advantage of the proposed idea over basic GCNs. Synthetic and results on QM7b QM9 are specific for triangles and cycles which is the model designed for. Overall, it is not very clear what the proposed idea brings to GNN in a general setting.

Given these concerns, I am leaning toward weak reject at this moment.


**Experience Assessment:**

I have read many papers in this area.

**Review Assessment: Checking Correctness Of Derivations And Theory:**

I did not assess the derivations or theory.

**Review Assessment: Checking Correctness Of Experiments:**

I assessed the sensibility of the experiments.

**Review Assessment: Thoroughness In Paper Reading:**

I read the paper at least twice and used my best judgement in assessing the paper.

---

> ### Author Response · Authors · 2019-11-05
> **Response to Review 3**
>
> Thank you Reviewer #3 for your review and comments. Here is an initial response:
>
> 1. We completely agree that Xu et al. 2019 did fundamental work in this area, and Theorem 3 is a natural extension of the work they did. Xu's original theorem seemed to implicate that GNNs aggregating over the immediate neighborhood are naturally constrained to 1-WL, and we show that by very so slightly changing the definition of the immediate neighborhood (L_1 arguably is still the immediate neighborhood) gives theoretically greater power. Moreover, the fact that it captures triangles leads to favorable performance in many areas.
>
> Furthermore, we believe Theorem 1 and Theorem 4 bring new contributions: specifically Theorem 1 formalizes what the N-GCN paper was proposing (look at larger aggregation regions) and shows that our hierarchy is complete under any aggregating regions that can be defined under the distance metric.
>
> Theorem 4 shows that our framework can, in the limit, discriminate between all graphs. Graph discrimination from a local point of view is hard (as from a local point one does not know if there are other disconnected segments), and the proof required the use of a special case of the Graph Reconstruction Conjecture.
>
> 2. We believe this would lead to the ability to better utilize node features in a more nontrivial/nonlinear way (as we can combine node features across greater regions) and should improve results for harder node classification tasks.
>
> 3. We agree that the baselines are not absolute state-of-the-art in each of these tasks. However, N-GCN and k-GNN represent the SOTA in moving away from immediate node aggregations, and their innovations are not orthogonal to other innovations (attention, residual, etc) in GNNs that would enable it to achieve better results. We chose not to include these to show clearly where the result improvement is coming from. Our results show that just by adding the L1 aggregation region, we are able to exceed results from N-GCN (which were missing the L-type aggregation regions, but theoretically looks at information up to 6 hops) and k-GNN (3 hops), demonstrating the importance of utilizing the hierarchy.
>
> The QM7b and QM9 datasets are extremely large chemical/biological molecule datasets which are prized as large real-world databases. This datasets were not specifically targeted for our purpose and the molecule dynamics are affected by many highly nonlinear, nontrivial, and nonlocal interactions between the atoms. Therefore, the fact that our models outperforms others in this regime is significant and potentially important for biological applications.

---

> > ### Author Response · Authors · 2019-11-09
> > **Paper Revised**
> >
> > Thank you Reviewer #3 for your comments. We have substantially revised the paper to address all your comments and uploaded the paper. We have addressed all of your other comments as below (labeling corresponds with your comments):
> >
> > 1. We agree our Theorem 3 is a natural extension of Xu's original theorem but Theorem 1 and 4 are new and has no equivalent in previous papers. We would kindly direct you to our initial response for detailed comments.
> >
> > 2. We have added a comment in the conclusion about our framework's ability to better utilize node features.
> >
> > 3(a). We have added significant number of new baselines to compare in our experimental results. We further removed the SOTA label throughout the paper with the understanding that it is contentious and potentially misleading.
> >
> > 3(b). We updated the results section to show clearly that our main comparison baselines are k-GNN and N-GCN. Since we have only changed the aggregation region from GCN, it is only fair that we compare to the previous best attempts from moving away from a local aggregation region, and our results show that we do outperform k-GNN and N-GCN significantly across datasets.

---

### Official Review · AnonReviewer2 · 2019-10-23
**Official Blind Review #2**

**Rating:** 8

**Review:**

The paper proposes minor modifications to Graph Convolutional Networks (GCNs) that as proven by the authors enable learning of local features in networks, namely the aggregation over powers of the adjacency matrix (effectively counting random walks within the neighborhood) and aggregating over connections within nodes in the neighborhood.

GCNs are of interest to
The paper is well written and clear.
The mathemeatical derivations are clearly structured.
The authors provide a large set of experiments on simulated and real data from different domains and relevant supervised tasks (node classifiation, graph classification and graph regression).
Source code is provided.

experimental results:
- For the simulation in Table 2, a 1-layer network seems like it would hield an advantage for the unstructured Erdös Renyi graphs. As I understand, the proposed GCN should by construction count the triangles (target variable) for each node embedding and then predict a linear function of those aggregated. My intuition says that this may even be an advantage over adding more layers. The text states the opposite. Here it would be interesting to see the effect of adding layers, or aggregating over longer random walks.
Here, it would be important to get the prediction of an untrained baseline in addition to the GCN baseline (e.g. the expected number of triangles/4cycles based on number of nodes in the Erdös Renyi model) to understand the scale the Mean Squared Error lives on and what the relative improvement means.

**Experience Assessment:**

I have published one or two papers in this area.

**Review Assessment: Checking Correctness Of Derivations And Theory:**

I assessed the sensibility of the derivations and theory.

**Review Assessment: Checking Correctness Of Experiments:**

I assessed the sensibility of the experiments.

**Review Assessment: Thoroughness In Paper Reading:**

I read the paper at least twice and used my best judgement in assessing the paper.

---

> ### Author Response · Authors · 2019-11-05
> **Response to Review 2**
>
> Thank you Reviewer #2 for your review and comments. Here is an initial response:
>
> We agree with your intuition that the deeper layers in this case might not be useful because the first layer should be able to count the triangles. We said we "handicapped" the models because deeper/wider models might be able to use higher-order features to reverse predict the number of triangles, which would make the comparison to GCN unfair.
>
> Our (incomplete) experiments when writing the paper suggest a larger aggregation area does improve the performance on these tasks but not by a lot. Thus we decided to keep our synthetic experiments to 1-layer so that it is also fair to the GCN (i.e. theoretically 3-layer GCN can look 3 hops away). Of course, if you feel like it is needed,  we are happy to add these results.
>
> We will add the prediction of an untrained baseline. The order is of 10^2 for the MSE of triangles, and 10^5 for the MSE of 4-cycles.

---

> > ### Author Response · Authors · 2019-11-09
> > **Paper Revised**
> >
> > Thank you Reviewer #2 for your comments. We have substantially revised the paper to address all of your comments and uploaded the paper. The specific changes corresponding to your comments are:
> >
> > 1. We added a baseline in which the model predicts the training mean onto the testing test. The results show that the GCN underperform such baseline significantly, while our modified networks are able to outperform the baseline significantly, further validating our results.
> >
> > 2. We have clarified the language surrounding "restricting" the GCN-L1 and GCN-D2 models to make it clear what we mean by "handicapping" the performance of these two models.

---

### Official Review · AnonReviewer1 · 2019-10-24
**Official Blind Review #1**

**Rating:** 3

**Review:**

This paper proposes a general class of GNN. The new model class generalizes the aggregation step to multiple levels of neighbors. The new model generalizes existing models. Theoretically, the paper shows the generalized models enjoy better discriminative power. The paper also conducts experiments to demonstrate the effectiveness of the new model class.


Comments:
1. The design of new models is straightforward, and the theoretical analysis is trivial, given Xu et al. 2019. The paper would be improved if the author(s) can provide optimization or generalization analysis.

2. The statement about experiments is misleading. The paper claims SOTA results on several datasets. However, this paper does not report recent SOTA results:
https://arxiv.org/abs/1809.02670
https://arxiv.org/abs/1810.00826
https://arxiv.org/abs/1905.13192

3. The use of the phrase "random walk" is weird. As there is no randomness at all.



**Experience Assessment:**

I have published one or two papers in this area.

**Review Assessment: Checking Correctness Of Derivations And Theory:**

I carefully checked the derivations and theory.

**Review Assessment: Checking Correctness Of Experiments:**

I carefully checked the experiments.

**Review Assessment: Thoroughness In Paper Reading:**

I read the paper thoroughly.

---

> ### Author Response · Authors · 2019-11-05
> **Response to Review 1**
>
> Thank you Reviewer #1 for your review and comments. Here is an initial response:
>
> 1. We completely agree that Xu et al. 2019 did fundamental work in this area, and Theorem 3 is a natural extension of the work they did. Xu's original theorem seemed to implicate that GNNs aggregating over the immediate neighborhood are naturally constrained to 1-WL, and we show that by very so slightly changing the definition of the immediate neighborhood (L_1 arguably is still the immediate neighborhood) gives theoretically greater power. Moreover, the fact that it captures triangles leads to favorable performance in many areas.
>
> Furthermore, we believe Theorem 1 and Theorem 4 bring new contributions: specifically Theorem 1 formalizes what the N-GCN paper was proposing (look at larger aggregation regions) and shows that our hierarchy is complete under any aggregating regions that can be defined under the distance metric.
>
> Theorem 4 shows that our framework can, in the limit, discriminate between all graphs. Graph discrimination from a local point of view is hard (as from a local point one does not know if there are other disconnected segments), and the proof required the use of a special case of the Graph Reconstruction Conjecture.
>
>
> 2. Thank you for pointing out these works. We are happy to add these works into our comparison. We would like to stress that the results in these papers do not significantly change what we presented. Specifically, it does not appear that any of the papers cited exceed the SOTA benchmarks cited in the paper (specifically the comparable datasets NCI1 (SOTA: 86.1), MUTAG (SOTA: 92.6)), Proteins (SOTA: 76.5), PTC-MR (SOTA: 63.6) ).  We do not claim we exceed these SOTA in our paper, and the results in this paper show this. We are happy to be more clear on this.
>
> Our example architectures performed the best on regression tasks (QM7b, QM9) where we were unable to find better results in the literature, but we were also prudent and did not claim we are setting the new SOTA.
>
> Further we would like to emphasize that this is a framework and the experiments are intended to show that powerful GNNs can be constructed in a straightforward way, exactly as you pointed out. This is by no means the "maximum" power available under this framework and a lot of what we proposed can be easily combined with the other innovations/advancements (attention, residual, etc) in the literature for even greater performance. The experiments intend to show that using our framework, we are able to achieve greater increase in performance compared to the GCN baseline than the previous SOTA attempts at doing so (N-GCN, k-GNN), despite looking at a smaller region (up to 2 hops) compared to k-GNN (3 hops) and N-GCN (6 hops). This shows the power of systematically adding features to a network.
>
> 3. The "randomness" refers to the fact that a length-k walk can be any random length-k walk. This follows terminology used in the N-GCN paper and other papers which relate random walks to GNNs. We are happy to update the terminology to the "set of all walks" for clarity.

---

> > ### Author Response · Authors · 2019-11-09
> > **Paper Revised**
> >
> > Thank you Reviewer #1 for your comments. We have substantially revised the paper to address all your comments and uploaded the paper. We have addressed all of your other comments as below (labeling corresponds with your comments):
> >
> > 1. We are unsure what you mean by optimization or generalization results as our proposed framework is general and not a specific algorithm. We believe our theoretical work is not trivial and would kindly direct you to our initial response for detailed comments.
> >
> > 2(a). We have removed references to SOTA in light of the understanding that this could be contentious/misleading.
> >
> > 2(b). We have added the results of all papers you mentioned into our article. We would like to stress that the new results do not fundamentally change our conclusion.
> >
> > 2(c). We updated the results section to show clearly that our main comparison baselines are k-GNN and N-GCN. Since we have only changed the aggregation region from GCN, it is only fair that we compare to the previous best attempts from moving away from a local aggregation region, and our results show that we do outperform k-GNN and N-GCN significantly across datasets.
> >
> > 3. We have removed the word "random" from our description of aggregation regions to make explicit that it is not random.

---

### Decision · Program_Chairs · 2019-12-19

**Decision:**

Reject

**Comment:**

This paper proposes a modification to GCNs that generalizes the aggregation step to multiple levels of neighbors, that in theory, the new class of models have better discriminative power. The main criticism raised is that there is lack of sufficient evidence to distinguish this works theoretical contribution from that of Xu et al. Two reviewers also pointed out the concerns around experiment results and suggested to includes more recent state of the art SOTA results. While authors disagree that the contributions of their work is incremental, reviewers concerns are good samples of the general readers of this paper— general readers may also read this paper as incremental. We highly encourage authors to take another cycle of edits to better distinguish their work from others before future submissions.